# Anthocyanins and Anthocyanidins in the Management of Osteoarthritis: A Scoping Review of Current Evidence

**DOI:** 10.3390/ph18030301

**Published:** 2025-02-21

**Authors:** Xiaodong Ma, Kok-Yong Chin, Sophia Ogechi Ekeuku

**Affiliations:** 1Department of Traditional Chinese Medicine, Universiti Tunku Abdul Rahman, Kajang 43000, Malaysia; mabo4388120@gmail.com; 2Department of Pharmacology, Faculty of Medicine, Universiti Kebangsaan Malaysia, Cheras 56000, Malaysia

**Keywords:** cartilage, inflammation, joint, osteoarthrosis, pigments

## Abstract

**Background/Objectives**: The consumption of food rich in anthocyanins, a natural pigment found in plants, has been associated with improved joint health. However, systematic efforts to summarise the effects of anthocyanins and their deglycosylated forms, anthocyanidins, in managing osteoarthritis (OA) are lacking. This scoping review aims to comprehensively summarise the current evidence regarding the role of anthocyanins and anthocyanidins in OA management and highlights potential research areas. **Methods**: A comprehensive literature search was performed using PubMed, Scopus, and Web of Science in January 2025 to look for primary studies published in English, with the main objective of investigating the chondroprotective effects of anthocyanins and anthocyanidins, regardless of their study designs. **Results**: The seven included studies showed that anthocyanins and anthocyanidins suppressed the activation of inflammatory signalling, upregulated sirtuin-6 (cyanidin only), and autophagy (delphinidin only) in chondrocytes challenged with various stimuli (interleukin-1β, oxidative stress, or advanced glycation products). Anthocyanins also preserved cartilage integrity and increased the pain threshold in animal models of OA. No clinical trial was found in this field, suggesting a translation gap. **Conclusions**: In conclusion, anthocyanins and anthocyanidins are potential chondroprotective agents, but more investigations are required to overcome the gap in clinical translation.

## 1. Introduction

Osteoarthritis (OA), a disease characterised by cartilage degeneration, subchondral osteophyte formation, and synovitis that manifests as joint pain, swelling, and stiffness [1], poises to impact global health significantly due to the increase in the ageing population. The global age-standardised prevalence rate of knee OA stood at 6967.29 per 100,000 [95% uncertainty interval (UI): 6180.7–7686.06] in 2021 [2]. The Global Burden of Diseases, Injuries, and Risk Factors Study (GBD) in 2021 assigned a disability-adjusted life-year value of 213 million (95% UI: 101.8–429.3 million) to OA, making it the 18th leading cause of disability [2].

Despite its prevalence, effective treatment for OA remains lacking, with pain relief using non-steroidal anti-inflammatory agents being the primary option. The effectiveness of symptomatic slow-acting drugs, such as glucosamine, chondroitin, and diacerein, is contentious so far. Viscosupplementation using hyaluronic acid offers merely temporary relief. Intra-articular glucocorticoid injections can relieve pain and inflammation, but their long-term safety remains a concern [3,4]. When all pharmacological agents fail, total knee replacement surgery is considered, but it comes with significant risks and potential side effects, such as thrombosis, infections, and periprosthetic fracture, which can complicate the recovery of the patients [5].

As a result, the exploration of potential compounds to mitigate OA progression, especially from nature, is gaining momentum [6]. Since inflammation and oxidative stress play key roles in the vicious cycle of cartilage destruction in OA [7,8], a compound with dual anti-inflammatory and antioxidant properties might have a role in mitigating the progression of OA. Many natural polyphenols, including flavones, stilbenes, and phenolic acids, have been investigated for their anti-OA effects, and the evidence has been extensively reviewed [8,9,10]. In contrast, reports on the anti-OA effects of anthocyanins, a class of flavonoid, are relatively scarce.

Anthocyanins, a group of hydrophilic pigments in the flavonoid class, are responsible for the red, purple, and blue colours found in various plants, such as berries, grapes, red cabbage, and eggplants. Anthocyanins are the glycosylated form of anthocyanidins (aglycones). The molecular structures of common anthocyanidins are depicted in Figure 1. Due to their attractive colours, anthocyanins help pollinate plants. They also protect plants from UV damage, drought, and pathogens [11,12]. Anthocyanins have been associated with many health benefits, such as protection from metabolic, cardiovascular, and neurodegenerative disorders as well as cancers [12,13].

The chondroprotective effects of anthocyanin are suggested by the benefits of red orange [14], purple corn [15], and purple rice extracts [16], which are rich in anthocyanins, in suppressing the progression of OA in preclinical models. Anthocyanins have been demonstrated to activate nuclear factor erythroid 2-related factor 2 (Nrf2) [17], the key regulator of the antioxidant response, which protects cells, including chondrocytes, from oxidative damage [18]. Anthocyanins also block the activation of nuclear factor kappa-B (NFκB) [19], the critical inflammatory pathway associated with various diseases, including OA [20]. More importantly, some studies have suggested that they stimulate the differentiation of mesenchymal stem cells into chondrocytes, which could enhance the regeneration of cartilage [21]. However, the anti-OA effects of anthocyanins have not been systematically reviewed.

Therefore, this scoping review aims to provide a comprehensive overview of the current evidence regarding the role of anthocyanins in OA management, highlighting their potential therapeutic benefits and identifying gaps in the existing research that warrant further investigation.

## 2. Methods

This scoping review was designed according to the steps outlined by Arksey and O’Malley [22], and adhered to the checklist of PRISMA for Scoping Reviews (Appendix A) [23]. The steps involved were (1) identifying the research question, (2) identifying relevant studies, (3) study selection, (4) charting the data, and (5) collating, summarising, and reporting the results. The protocol of this scoping review is available at Open Science Framework (Url: https://osf.io/8xjrk/?view_only=181e036019cf46b5ad6d58825a7381b0, assessed on 20 January 2025).

### 2.1. Identifying the Research Question

The current scoping review addressed the question, “What are the effects of anthocyanin and anthocyanidin supplementation on OA?” The research question of this review was designed based on the Population, Concept, and Context (PCC) outlined in Table 1. The research question was addressed with evidence derived from studies using preclinical models of OA (in vitro or in vivo) and patients with OA (clinical trials) treated with anthocyanins or anthocyanidins.

### 2.2. Identifying Relevant Studies

In January 2025, a systematic literature search was performed on three major scholarly databases, i.e., PubMed, Scopus, and Web of Science, using the search string (Anthocyanin* OR Anthocyanidin* OR Aurantinidin OR Capensinidin OR Cyanidin OR Delphinidin OR Europinidin OR Hirsutidin OR Malvidin OR Pelargonidin OR Peonidin OR Petunidin OR Pulchellidin OR Rosinidin) AND (Osteoarthritis OR Osteoarthrosis OR Cartilage OR Chondrocyte*). The search string was applied to titles and abstracts to avoid unspecific results. All items between the inception of databases and the date of search were included. No additional filter was applied during the search.

Primary studies published in English with the main objective of studying the effects of anthocyanin or anthocyanidin supplementation on joint health parameters in OA models or patients were included. Items without primary data, such as editorials, letters, reviews (including systematic reviews and meta-analyses), and perspectives, were excluded. Conference abstracts and proceedings were excluded due to incomplete data and duplication with full articles. Studies investigating the effects of anthocyanin OR anthocyanidin exposure on normal chondrocytes or animals without OA induction were excluded. Studies using crude extracts or mixed formulations were excluded because the effects of anthocyanins OR anthocyanidins could not be delineated.

### 2.3. Study Selection

Reference management was performed using Endnote version 20.6 (Clarivate, Philadelphia, PA, USA). The search results from three databases were merged and deduplicated using Endnote. A manual examination was performed to ensure the deduplication was successful. The titles and abstracts were screened by two researchers (X.M., K.-Y.C.) independently based on the inclusion and exclusion criteria. Next, the full texts were obtained for the eligible items and screened by the same researchers. Disagreements were resolved by discussion and the opinions of the third researcher (S.O.E.). The reference list of the included articles and relevant reviews was screened to identify potential articles left out during the literature search.

### 2.4. Charting the Data

Data from the articles included was extracted by two researchers (X.M., K.-Y.C.) using a standardised Google Form (Google, Mountain View, CA, USA). The data extracted included authors, year of publication, study designs, OA models adopted, treatment details (compounds, dosages, and treatment periods), major findings, limitations, and conclusions.

### 2.5. Collating, Summarising, and Reporting the Results

The data were summarised and reported qualitatively based on important pathological OA features, such as chondrocyte viability, anabolic and catabolic balance of cartilage, cartilage integrity, and pain and behaviour changes. Data synthesis based on statistical methods was not conducted because the study designs, OA models, and reported outcomes were heterogeneous and difficult to combine meaningfully. The role of anthocyanins and anthocyanidins in mitigating OA in each cascade of the pathogenesis of the disease, current research gaps in the field, and limitations of the review are covered in Section 4.

## 3. Results

The literature search uncovered 44 unique articles from three scholarly databases, of which 37 were excluded because they were not within the scope (*n* = 12), did not contain primary data (*n* = 14), did not use OA models (*n* = 6), used crude extracts (*n* = 4), or were conference abstracts (*n* = 1). Seven articles were included in the current scoping review. The results of the article screening and selection are summarised in Figure 2.

### 3.1. Study Characteristics

Four studies used in vitro models, and three studies used a combination of in vitro and in vivo models to examine the effects of anthocyanins and anthocyanidins in OA models. No clinical studies have been conducted on this topic. For in vitro studies, primary chondrocytes of humans (normal [15,16] and OA patients [24,25]) and mice [26,27], as well as human chondrocyte cell line C28/I2 [28], have been used. Most studies used interleukin-1β (IL-1β) to mimic OA changes in in vitro studies [16,24,25,26,27], whereas a study used advanced glycation products to mimic OA changes in diabetic conditions [15], and one study used hydrogen peroxide to emulate oxidative stress in OA. For in vivo studies, C57B/L male mice [25,26] and Wistar rats (sex not specified) [27] were used. The OA induction methods were medial meniscus destabilisation (MMD) in mice [25,26] and monosodium iodoacetate (MIA) in Wistar rats [27].

The compounds investigated included cyanidin, delphinidin, malvidin, peonidin, pelargonidin, and their corresponding aglycone forms. In in vitro studies, various dosages (1.25–50 µM) and treatment periods were attempted depending on the objectives, while in vivo studies used oral dosages ranging from 5 to 20 mg/kg/day for 14 days to 8 weeks.

### 3.2. Effects on Chondrocyte Viability

Chondrocytes undergo apoptosis when exposed to stressors, such as IL-1β and oxidative stress [27,29]. Since chondrocytes are the only cell type in cartilage to synthesise the extracellular matrix (ECM), the capacity for cartilage to regenerate decreases, leading to OA [30].

Treatment with anthocyanins prevented the apoptosis of chondrocytes [27,28]. For instance, malvidin reduced the number of chondrocytes in rats expressing senescence-associated B-galactosidase after IL-1β exposure. This effect was attributed to the suppression of nuclear-factor kappa-B activation [27]. Delphinidin reduced the terminal deoxynucleotidyl transferase dUTP nick-end labelled C28/I2 chondrocytes after hydrogen peroxide exposure. This reduction was accompanied by a suppression of pro-apoptotic markers [cleaved caspase-3 and cleaved poly (ADP-ribose) polymerase] and an increase in anti-apoptotic markers (B-cell lymphoma-extra large) [28]. This effect was associated with the activation of autophagy, marked by increased microtubule-associated protein 1A/1B-light chain 3 expression and autophagic vacuoles [28].

### 3.3. Effects on Anabolic and Catabolic Processes of Chondrocytes

The anabolic process in chondrocytes is reflected by the expression of ECM components, such as aggrecan and type II collagen. Meanwhile, the catabolic process in chondrocytes is reflected by the expression of metalloproteinases (MMPs) responsible for breaking down the ECM [31]. In OA, inflammation skewed the balance between anabolic and catabolic processes to the latter, contributing to cartilage degradation [32].

Various in vitro and in vivo studies have showcased the ability of anthocyanins and anthocyanidins to promote the anabolic process and prevent the catabolic process of chondrocytes. Notably, pelargonidin-treated primary mouse chondrocytes stained strongly for its proteoglycan content, and showed reduced disintegrin and metalloproteinase with thrombospondin motifs 5 (ADAMSTS5) and MMP-13 expressions [26]. Similarly, in mice with MMD-induced OA, pelargonidin treatment (10–20 mg/kg/d for 8 weeks) increased aggrecan expression while decreasing MMP-13 expression at the cartilage [26].

In another study, peonidin-3-glycoside and its deglycosylated form, as well as cyanidin-3-glycoside but not its deglycosylated form, reduced MMP-1, -3, and -13 mRNA expression [16]. The regulatory effects of cyanidin were further validated in another study, whereby cyanidin-treated chondrocytes from OA patients exposed to IL-1β maintained the protein expression of aggrecan, type II collagen, and SRY-box transcription factor 9, while reducing the expression of MMP-13 and ADAMTS4. These effects are mediated by sirtuin-6 [25].

### 3.4. Effects of Cartilage Integrity

Cartilage integrity is the ultimate measure of joint health because it represents the net effects of treatment. In animal studies, the joint can be harvested and sectioned, and the cartilage morphology can be evaluated under a microscope and scored [33,34]. Surprisingly, only one included study used the Osteoarthritis Research Society International (OARSI) Score to assess cartilage integrity in rats with MMD-induced OA. The OARSI Score reduced after pelargonidin supplementations (20–30 mg/kg/day for 8 weeks) [26].

### 3.5. Effects on Pain Threshold

OA is manifested as joint pain in animals and humans with OA [35]. MIA-induced OA is an excellent model for evaluating pain response [36]. Malvidin (10 and 20 mg/kg/day for 14 days) increased the paw pressure and joint compression threshold of rats with MIA-induced OA, indicating greater pain tolerance [27].

### 3.6. Mechanism of Actions

Most of the studies attributed the joint benefits of anthocyanins and anthocyanidins to their anti-inflammatory effects, particularly through the suppression of NFκB activation [15,16,24,25,27]. The nuclear translocation of NFκB or p65 is required for the translation of pro-inflammatory genes [37]. Treatment with anthocyanins reduced the phosphorylation and nuclear translation of p65 in chondrocytes. However, the mechanisms of different anthocyanins and anthocyanidins could be different. A study illustrated that cyanidin, peonidin, and their deglycosylated forms suppressed the activation of the IκB kinase complex and I-kappa-B related to the canonical NFκB pathway in human primary chondrocytes exposed to IL-1β, but only the glycosylated forms prevent the activation of p-65 [16]. Only peonidin, among the four compounds tested, can activate JNK pathway. None are effective in activating p-38 [16]. JNK and p38, along with ERK, form MAPK signalling, which is also vital to the inflammation process in OA [38]. However, in AGE-exposed human primary chondrocytes, cyanidin-, peonidin-, and malvidin-3-glycoside could suppress the activation of IKK, IκB, p65, ERK, p-38, and JNK altogether [39]. The model difference could be responsible for the discrepancy in the molecular effects of anthocyanins.

Autophagy is another cellular mechanism critical for maintaining chondrocytes’ health [40]. Chloroquine, an autophagy inhibitor, was shown to ameliorate the anti-apoptotic effects of delphinidin in human chondrocyte C28/I2 exposed to hydrogen peroxide [28]. Sirtuin-6, an anti-senescence molecule, also plays a role in mediating the chondroprotective effects of anthocyanins. Sirtuin-positive cells increased in mice with MMD-induced OA treated with 5 mg/kg/day cyanidins for 8 weeks [25]. Cyanidin also improved the expression of sirtuin-6 in primary human chondrocytes exposed to IL-1β, but its silencing abolished the chondroprotective effects of cyanidins [25].

The study designs and major findings of the included studies are summarised in Table 2.

## 4. Discussion

The pathogenesis of OA is multifaceted, involving not only mechanical wear and tear but also biochemical signalling that exacerbates synovial inflammation and cartilage degradation [41]. The mechanical wear and tear of cartilage generates particles known as damage-associated molecular patterns, which bind with Toll-like receptors (TLRs) on the surface of invading macrophages, chondrocytes, and synoviocytes. This binding recruits adapters with TLR-IL1 domains to initiate downstream signalling cascades, which include kinases such as MAPK and IKKs, both of which regulate the phosphorylation of transcription factors important in inflammation [42]. Pro-inflammatory cytokines stimulate chondrocytes and synoviotes to produce MMPs such as MMP-1, MMP-3, and MMP-13, degrading collagen and proteoglycans, leading to cartilage destruction [43]. As evidenced by previous studies, anthocyanins are able to suppress inflammation via the regulation of MAPK and NFκB signalling and reduce the cartilage degradation process by downregulating MMP expression [15,16].

In OA, chondrocytes, the sole cell type in cartilage, increase ECM production in an attempt to repair the damaged cartilage. The increased workload drives the hypertrophy of chondrocytes. They eventually undergo premature senescence, loss of functions, and apoptosis [44]. Sirtuin-6 is known to play a role in DNA repair and preserving mitochondrial functions in chondrocytes [45]. Malvidin was shown to downregulate β-galactosidase, a cellular senescence marker, in the cartilage of mice with OA [27]. In addition, as an activator of sirtuin-6, cyanidin preserved anabolic processes while reducing catabolic processes in chondrocytes in OA conditions [25]. Interestingly, cyanidin also prevents chondrogenic and hypertrophic differentiation of mesenchymal stem cells by blocking autophagy [46]. This observation warrants the examination of the effects of cyanidins on hypertrophic chondrocytes in OA conditions.

Autophagy is an important process of chondrocyte homeostasis, as it prevents the accumulation of damaged organelles that might negatively impact the functions of chondrocytes, eventually leading to apoptosis [47]. Autophagy is activated during early OA as a protective mechanism but is downregulated as the disease progresses. Thus, the delicate stimulation of the autophagy process could mitigate the progression of OA [48]. Delphinidin could enhance autophagy in chondrocytes assaulted with oxidative damage and prevent their apoptosis. Blocking autophagy could nullify the benefits of delphinidin, signifying the central role of this process in chondroprotection [28].

The enhanced anabolism and reduced catabolism of cartilage could explain the preservation of cartilage integrity in mice with OA treated with pelargonidin. On the other hand, the increased pain threshold, as observed in malvidin-treated rats with OA, could be a result of improvement in joint structure (although premature to be concluded in a short-term study as in [27]), but is more likely a result of the suppression of the inflammation important in exacerbating pain transduction. Inflammatory mediators may activate peripheral nociceptors by stimulating the further release of inflammatory mediators and sensitising primary afferent neurons to other stimuli [49].

The mechanism of anthocyanins and anthocyanidins in managing OA is summarised in Figure 3.

Though not explored in the current review, anthocyanins and anthocyanidins could manage OA by indirect mechanisms. Firstly, anthocyanins have been demonstrated to mitigate metabolic syndromes and obesity in many studies [50]. Since obesity is a significant risk factor for OA due to increased mechanical loading on the joint, systemic inflammation and oxidative stress [51], anthocyanins and anthocyanidins could relieve the effects of obesity on the joint. Secondly, previous studies have demonstrated that anti-osteoporosis medications can help improve joint health in patients with OA [52]. This benefit may be a consequence of enhanced subchondral structures, which transfer less loading stress to the cartilage [53]. The intake of anthocyanin-rich food has been associated with improved bone health in middle-aged and older adults at risk of osteoporosis, according to a meta-analysis [54]. Preclinical studies have demonstrated the potential pro-osteogenic and anti-resorptive effects of anthocyanins [55]. Thus, anthocyanins could preserve subchondral bone and cartilage integrity at the same time. Thirdly, the bioavailability of anthocyanins is low after oral consumption [56]. They are metabolised to other phenolic compounds like protocatechuic acid (PCA) [57], with potent chondroprotective effects. A study found that PCA deterred the loss of ECM components from cartilage explants in culture and showed shared anti-inflammatory mechanisms with anthocyanins [16]. It has also been incorporated into biomaterials for OA treatment [58]. Further studies should be conducted to examine whether these metabolites are responsible for the chondroprotective effects of anthocyanins. Lastly, there is a surge of interest in the gut–joint axis, whereby the relationship between gut microbiota and OA is mediated by inflammation, oxidative stress, and metabolic dysfunction [59,60]. Anthocyanins have been shown to regulate gut microbiota composition in achieving its therapeutic potentials [61,62,63]; therefore, their chondroprotective effects might be mediated by the gut–joint axis.

Several limitations were noted from the studies included in the current review. For the in vitro studies, the replication of findings in more than one chondrocyte source should provide stronger evidence on the effects of anthocyanins and anthocyanidins. The upstream mechanism leading to the activation of MAPKs and IKK, such as tumour necrosis factor receptor-associated factor 6 (TRAF6) and myeloid differentiation primary response 88 (MyD88), which are central to various inflammatory pathways [64], have yet to be explored. Similarly, the regulatory mechanisms of sirtuin-6, autophagy, and Nrf2 by anthocyanins and anthocyanidins are not precise as of current and warrant further investigation. Cartilage explants provide a simple three-dimensional culture system to observe the effects of xenobiotics on cartilage metabolism, such as ECM component release and retention. However, this type of experiment has only been attempted for PCA, as well as anthocyanin-rich food extracts [15,16], but not anthocyanins or anthocyanidins. For the in vivo studies, only one explored the effects of anthocyanins on cartilage integrity, and one examined the effects of anthocyanins on pain response. For clinical translation, the preservation of cartilage tissue and the amelioration of symptoms are important and should be emphasised in the future. Most importantly, none of the vivo studies answered whether anthocyanins, anthocyanidins, or their metabolites could enter the circulation and subsequently deposit in the joint space to exert their effects. The apparent lack of clinical trials in the field should be of concern because the effects of anthocyanins or anthocyanidins have not been verified in humans. Our research in clinicaltrial.gov dated January 2025 revealed no ongoing or completed trials on this topic. This gap in clinical translation should be bridged in the future.

The current scoping review is not without its limitations. We only searched for items published in the English language in three major scholarly databases. Thereby, selection bias could not be avoided due to exclusion of studies published in other languages and not indexed by these databases. We did not search for grey literature, so there might be a bias in including studies reporting favourable outcomes. We also did not conduct quality assessments of the studies included. However, an appraisal of possible limitations of the studies is mentioned in Table 2. Nevertheless, this scoping review provides the current landscape of the research on the chondroprotective effects of anthocyanins or anthocyanidins and suggests possible future studies for this field.

## 5. Conclusions

Anthocyanins and anthocyanidins exert chondroprotective effects via the modulation of inflammation signalling pathways (MAPK and NFκB) in chondrocytes. Notably, cyanidin also upregulates sirtuin-6, and delphinidin upregulates autophagy in chondrocytes. Specific types of anthocyanins, like malvidin, have been found to be effective in abolishing inflammatory pain, and pelargonidin has been found to preserve cartilage integrity. A translation gap exists to date as no clinical trials on this topic can be found. Future research efforts should be geared towards this direction to verify the effectiveness of anthocyanins and anthocyanidins in managing OA.

## Figures and Tables

**Figure 1 pharmaceuticals-18-00301-f001:**
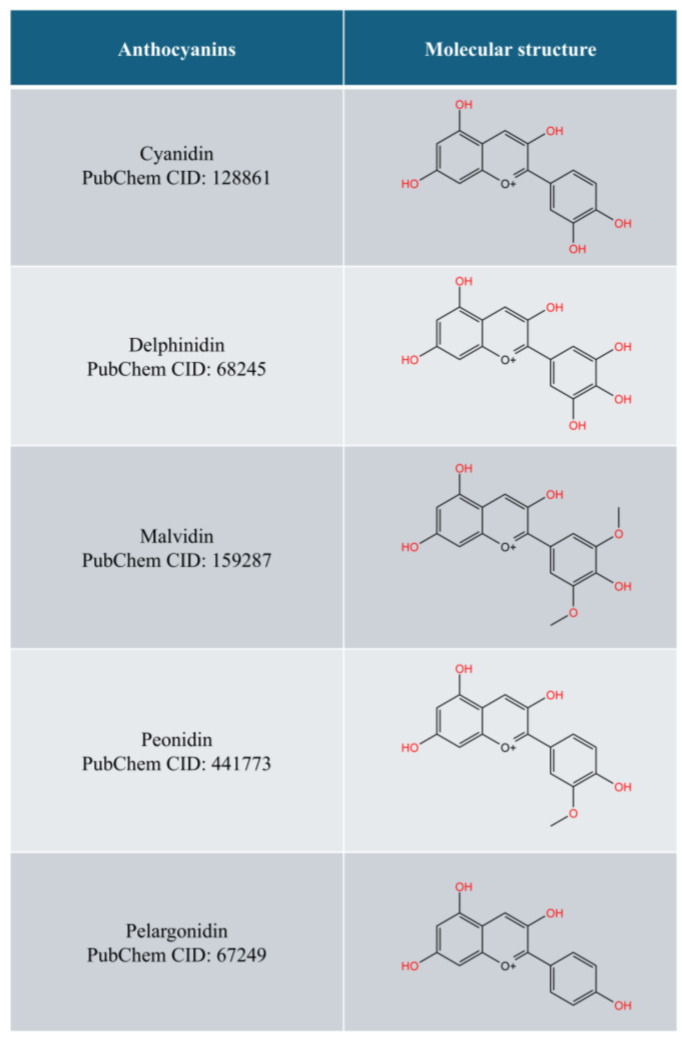
The common forms of anthocyanidins (drawn with KingDraw).

**Figure 2 pharmaceuticals-18-00301-f002:**
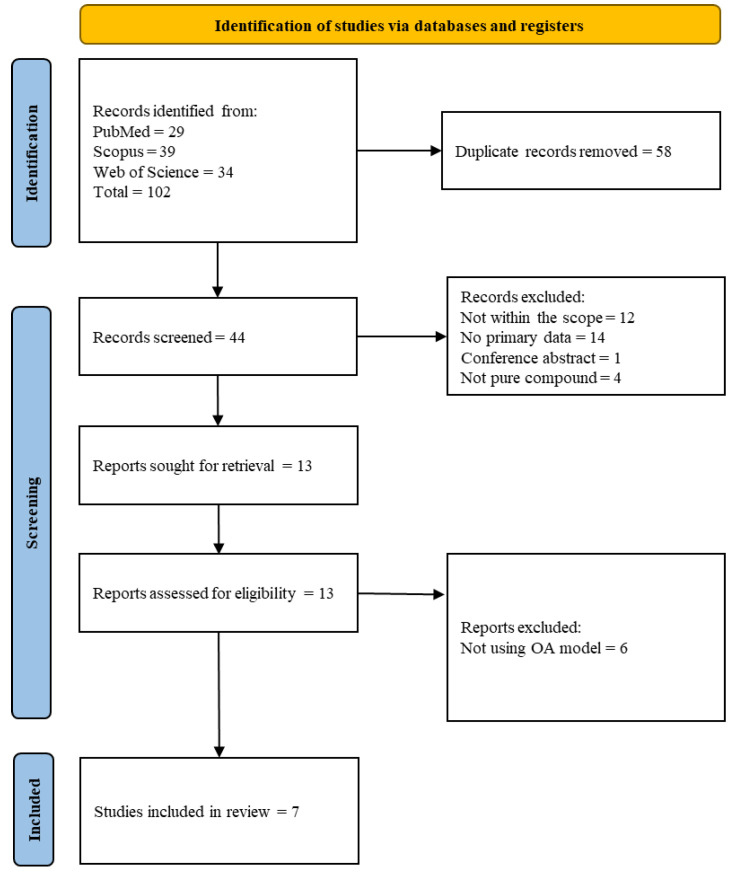
PRISMA flow chart (adapted from https://www.prisma-statement.org/prisma-2020-flow-diagram, assessed on 20 January 2025).

**Figure 3 pharmaceuticals-18-00301-f003:**
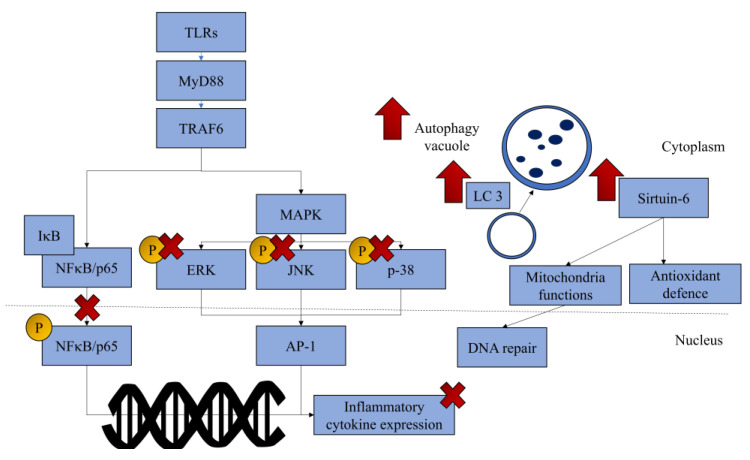
The regulatory mechanism of anthocyanins and anthocyanidins in mitigating the progression of OA. Most anthocyanins and anthocyanidins can prevent inflammation and specific anthocyanins, such as cyanidin, which has been reported to upregulate sirtuin-6, whereas delphinidin has been reported to upregulate autophagy. (Drawn with Microsoft PowerPoint.) Abbreviations: AP-1, AP-1 transcription factor; ERK, extracellular signal-regulated kinase; IκB, I-kappa-B; JNK, c-Jun N-terminal kinase; LC3, microtubule-associated protein 1A/1B-light chain 3; MAPK, mitogen-activated protein kinase; NFκB, nuclear factor kappa-B; TRAF6, tumour necrosis factor receptor-associated factor 6; up arrow (↑), upregulation; cross symbol (×), inhibition.

**Table 1 pharmaceuticals-18-00301-t001:** The Population, Concept and Context of the current scoping review.

Concept	Description
Population	Chondrocytes induced with OA changes Animals induced with OA Patients with OA
Concept	Anthocyanins or anthocyanidins
Context	In vitro, in vivo, or clinical trials

**Table 2 pharmaceuticals-18-00301-t002:** Summary of published studies on the effects of anthocyanins and anthocyanidins on OA.

Authors (Year)	Study Type	OA Model/Subject Characteristics	Treatments	Parameters Increased vs. OA Control	Parameters Decreased vs. OA Control	Parameters Unchanged vs. OA Control	Limitations	Remarks
Haseeb et al. (2013) [24]	In vitro	Chondrocytes from OA patients’ femoral head undergoing hip replacement surgery, induced with IL-1β (1–5 ng/mL for inflammatory assays, 10 ng/mL for NFκB signalling).	Treatment: delphinidin (10 µg/mL for inflammatory assays, 50 µg/mL for NFκB signalling) first for 2 h, followed by IL-1β exposure (24 h for inflammation assays, 30 min for NFκB signalling). OA control: IL-1β exposure, no treatment. Positive control: BAY 11-7082 (2.5 mM for inflammatory assays, 10 mM for NFκB signalling).	NA	Inflammation markers: COX-2 mRNA and protein expressions, PGE2 production NFκB signalling: p-IRAK-1Ser376, p-IKKα/β, IKKβ mRNA and protein expression, IκBα degradation, NFκB activation and nuclear translocation, p-NF-κB-inducing kinase	p-TGF-β-activated kinase 1	Anabolic and catabolic markers of chondrocytes not studied.	Delphinidin inhibits COX-2 expression and PGE2 by suppressing the activation of NFκB signalling by preventing the phosphorylation of IRAK-1Ser376
Dai et al. (2017) [27]	In vitro, in vivo	In vivo: Wistar rats (8–12 weeks, sex not specified), injected with MIA (2 mg/kg) intra-articularly. In vitro: Apoptosis and inflammation assays: Chondrocytes from sham and OA group of the in vivo study. NFκB signalling: Chondrocytes from the femoral head of C57BL/6 mice exposed to IL-1β. Luciferase assay: 293T cells transfected with IjBa-specific siRNA.	In vivo: Treatment: malvidin (p.o., 10, 20 mg/kg/day) for 14 days. OA control: MIA injection without treatment (2 mg/kg). Positive control: NA.	In vivo: Day 14: Paw pressure and joint compression threshold (reduction in pain).	In vitro: Day 14: Apoptosis: Senescence-associated β-galactosidase-stained chondrocytes Days 1 and 7: Inflammation: Expression of IL-1β, IL-6, TNF-α, MMP3, MMP9, MMP13 NFκB: p65 nuclear translocation	In vitro: NFκB signalling: p-IKKα/β, IKKβ protein expression Luciferase assay: silencing of IκBα did not affect the effects of malvidin	Cartilage degradation not illustrated	Malvidin prevents pain and inflammation in rats with OA by suppressing the activation of NFκB signalling, but its mechanism is independent of IκBα
Jiang et al. (2019) [25]	In vitro, in vivo	In vitro: OA patients’ primary chondrocytes undergoing total knee replacement, exposed to IL-1β (10 ng/mL). In vivo: Animals: 10-week-old C57BL/6 male mice. OA induction: MMD.	In vitro: Co-incubation of chondrocytes with cyanidin (12.5, 25, and 50 µM). In vivo: Treatment: 5 mg/kg/day cyanidin for 8 weeks, oral. Negative control: normal saline. Positive control: NA.	In vitro: The protein expression of aggrecan, collagen II, SOX9 (12.5–50 µM, except collagen II at 25 µM). The protein expression and fluorescence intensity of sirtuin 6 (25–50 µM). * Silencing sirtuin 6 abolished all these changes. In vivo: Collagen II- and sirtuin 6-positive cells.	In vitro: Inflammation: mRNA and protein expression of iNOS, COX2, TNF-α, IL-6, the levels of NO and PGE2 (25 and 50 µM) Cartilage degradation markers: protein expression of MMP-13 and ADAMST4 (25 and 50 µM) NFκB signalling: phosphorylation of p65 and pIκB * Silencing sirtuin 6 abolished all these changes In vivo: OARSI scores, synovitis scores MMP-13 positive cells in the cartilage	NA	The anti-inflammatory effects were not replicated in the in vivo model	Cyanidin activates sirtuin 6 to suppress inflammation and achieve its chondroprotective effects
Lee et al. (2020) [28]	In vitro	C28/I2 human chondrocyte cells exposed to hydrogen peroxide (500 µM).	Delphinidin (40 µM). Negative control: not treated. Positive control: NAC (5 mM).	Cell viability Anti-apoptosis proteins: Bcl-Xl levels. Antioxidant response proteins: Nfr2, p-NFκB. Autophagy markers: LC3 expression, autophagic vacuoles (staining) [increased by H_2_O_2_, further increased in the presence of delphinidin]. * These changes were inversed in the presence of chloroquine, an autophagy inhibitor.	Pro-apoptosis markers: c-caspase-3 and c-PARP Apoptotic cells (TUNEL assay)	NA	Not human primary cell line Upstream mechanism of autophagy not investigated	Delphinidin protects chondrocytes subjected to oxidative stress by activating Nrf2, NFκB, and autophagy of the cells.
Chuntakaruk et al. (2021) [15]	In vitro studies	Porcine cartilage exposed to AGE (25 μg/mL). Non-OA patients’ joint material, exposed to AGE (10 μg/mL).	Purple corn anthocyanins, containing cyanidin-3-O-glucoside chloride (C3G, 53.39 ± 0.54 μg/g of crude extract), pelargonidin-3-O-glucoside (P3G, 34.21 ± 0.13 μg/g of crude extract), peonidin-3-O-glucoside chloride (P3GC, 33.18 ± 0.12 μg/g of crude extract), and malvidin-3-O-glucoside (M3OG, 16.50 ± 0.05 μg/g of crude extract): 6.25–25 μg/mL for explant assays for 35 days. C3G: 1.25–5 μM; P3G: 2.5–10 μM; P3GC: 2.5–10 μM; Protocatechuic acid (PCA): 2.5–10 μM.	Retention of uronic acid in the explant: PCA. Glycosaminoglycan retention in the explant (Safranin O staining): PCA.	Glycosaminoglycan and hyaluronic acid release from cartilage explant: PCA mRNA expression of MMP1, 3, 13: PCA and all constituents p-IKK/IKK, p-IκB/IκB, p-p65/p65: PCA and all constituents p-ERK/ERK, p-p-38/p38, p-JNK/JNK: PCA and all constituents	Chondrocyte morphology (H&E staining) P3G not effective on p-p38	Unclear details on human cartilage donor AGE-RAGE signalling study not performed	Purple corn anthocyanin and its constituents protect against cartilage loss due to AGE through NFκB and MAPK signalling pathways
Wongwichai et al. (2019) [16]	In vitro studies	Explant study: Porcine articular cartilage exposed to IL-1β (25 ng/mL for 3 days, 10 ng/mL for 35 days in the presence of oncostatin M 10 ng/mL). Chondrocyte study: Non-OA human articular chondrocytes, exposed to IL-1β (2 ng/mL, duration varies on experiments).	PCA: Explant study: 6.25–50 µg/mL; chondrocyte study: 2.5–10 µM. Anthocyanidins: - Cyanidin-3-O-glucoside chloride (C3G): 2.5–10 µM. Peonidin-3-O-glucoside chloride (P3G): 2.5–10 µM. Anthocyanins: Cyanidin chloride (CC): 2.5–10 µM; Peonidin chloride (PC): 2.5–10 µM; PCA: 2.5–10 µM.	NA	Explant study: PCA: release of glycosaminoglycan (7–14 days), hyaluronic acid (7–28 days), hydroxyproline (21–35 days) Chondrocyte study: mRNA expression of MMP-1, MMP-3, MMP-13: C3G, P3G, PC, PCA p-IKK/IKK, p-IκB/IκB: C3G, P3G, CC, PC, PCA p-p65/p65: C3G, P3G, PCA p-ERK/ERK: C3G, P3G, PC, PA p-JNK/JNK: PC	Explant study: PCA: Uronic acid retention, cartilage and chondrocyte morphology (H&E), glycosaminoglycan retention (Safranin-O staining), collagen retention (staining) p-p65/p65: CC, PC p-p38/p38: C3G, P3G, CC, PC, PCA p-JNK/JNK: C3G, P3G, CC, PCA	Unclear details on human cartilage donor	Anthocyanidins inhibit NFκB activation more effectively than anthocyanins in preventing cartilage degradation
Zeng et al. (2023) [26]	In vitro, in vivo	In vitro: Primary chondrocytes from neonatal mice’s knee cartilage, exposed to IL-1β (10 mg/mL). In vivo: Medial meniscus destabilisation in C57BL/6 mice.	In vitro: pelargonidin (10–40 μM) for 24–72 h. In vivo: pelargonidin (10 mg/kg/d, 20 mg/kg/d) for 8 weeks. Negative control: normal saline. Positive control: NA.	In vitro: Toluidine blue staining, collagen II protein expression. In vivo: Aggrecan expression (IHC).	In vitro: mRNA expression of IL-6, TNF-α, COX-2 and iNOS; protein expression of COX-2, iNOS, ADAMTS5 and MMP13 p-p65/p65 expression and nuclear translocation of p65 In vivo: OARSI score (Safranin O) MMP13 expression (IHC)	NA	Single cell line, not from human Inflammatory effects not demonstrated in vivo	Pelargonidin prevents cartilage degradation by suppressing NFκB activation

Abbreviations: ADAMTS, A Disintegrin and Metalloproteinase with thrombospondin motifs; Bcl-Xl, B-cell lymphoma—extra large; COX-2, cyclooxygenase-2; ERK, extracellular signal-regulated kinase; H&E, haematoxylin and eosin; IHC, immunohistochemistry; IκB, I-kappa-B; IKK, I kappa B kinase; IL-1β, interleukin-1β; iNOS, inducible nitric oxide synthase; IRAK, interleukin-1 receptor-associated kinase; JNK, c-Jun N-terminal kinase; LC3, microtubule-associated protein 1A/1B-light chain 3; MIA, monosodium iodoacetate; MMD, medial meniscus destabilisation; MMP, metalloproteinase; NA, not available; NAC, N-acetylcysteine; NFκB, nuclear factor kappa-B; NO, nitric oxide; Nrf2, nuclear factor erythroid 2-related factor 2; OA, osteoarthritis; OARSI, Osteoarthritis Research Society International; P-, phosphorylated; PGE2, prostaglandin E2; TGF, transforming growth factor; TNF; tumour necrosis factor; TUNEL, terminal deoxynucleotidyl transferase-mediated dUTP nick-end labelling; *, the use of antagonist.

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
