# Peer review of "Anthocyanins and Anthocyanidins in the Management of Osteoarthritis: A Scoping Review of Current Evidence"

_pharmaceuticals, 2025, doi:10.3390/ph18030301_

Round 1

Reviewer 1 Report

Comments and Suggestions for Authors

The scooping review article "Anthocyanins and anthocyanidins in the management of osteoarthritis: A scoping review of current evidence" by Xiaodong Ma et al., systematically identifies and maps the currently limited available evidence of chondroprotective effects of anthocyanins and anthocyanidins and their possible mechanisms of action. The review adheres to PRISMA checklist (which is provided in the supplementary table S1), based on 7 literature sources after thorough filtering of 58 deduplicated publications on the topic. The The information included in this review is well-organized, presented in a well-structured manner and easy to read. The choice of scooping type over systematic review is clearly defined, as the article tries to identify the types of chondroprotective effects of the given substances from a number of articles highlighting possible limitation of their methodology. I have only a few small suggestions:

1. Figure 1, the fourth row contains the PubChem CID and a molecular structure for glycosylated version of peonidin - Peonidin-3-O-glucoside. Please check whether peonidin or peonidin-3-O-glucoside was intended and change either text or molecular structure accordingly.

2. Table 2, page 12. Last row on this page - " PCA Glycosaminoglycan retention in the explain (Safranin O staining): PCA". Probably there is a typo and the "explant" was intended.

3. Page 18, figure 3. The choice of dark blue colour and black text impedes the readablility. I advice to use a lighter background colour.

4. Page 18, lines 56-60. I think abbreviations are better to be included in the figure 3 description, than in the main text.

I think the article is of interest to the readers. The quality of review is convincing, only some minor problems needs to be addressed (minor revision) before it can be accepted for publication.

Author Response

Dear reviewer,

Thank you for the meticulous review. Please see the attachment for our response. Thank you.

Reviewer 2 Report

Comments and Suggestions for Authors

Paper entitled Anthocyanins and Anthocyanidins in the Management of Osteoarthritis: A Scoping Review of Current Evidence” by Xiaodong Ma, Kok-Yong Chin,* and Sophia Ogechi Ekeuku brings an overview of the research on the chondroprotective effects of anthocyanins / anthocyanidins and suggests possible future studies in this field.

Comments

1.       The first sentence in the Introduction is unclearly composed. It should be checked and re-composed.

2.       Line 52:  flavones should be changed to flavonoids.

3.       Figure 1 shows only anthocyanidins (aglycones). This should be stated/changed in the Introduction (lines 56 and 57) or glycosides (anthocyanins) and their corresponding aglycones should be added to Figure 1. The description of Figure 1 should also be adjusted accordingly.

4.       Section 2.1. The need and meaning of Table 1 are unclear. It should be clarified.

5.       The terms in vitro and in vivo should be written in italics. This should be changed throughout the text.

6.       The studies included in the review are presented and discussed in detail. However, it seems that the exclusion factors were very rigorous and therefore a large number of the considered studies were discarded. Although the authors provided a short explanation (e.g., Not within the scope), the question is whether the review paper could have included any of the excluded studies?

7.       In Figure 3, a different colour should be applied to the rectangles (e.g. light blue instead of dark blue because the text is not visible).

8.       The description of the included studies is correctly arranged, and some parts are well-composed (especially tables summarizing the results of individual subject studies (under Table 2); the suggestion is to put the tables in landscape form and additionally shorten individual descriptions (it is summary of published studies).

Author Response

(The authors gave the same response as above.)

Reviewer 3 Report

Comments and Suggestions for Authors

The manuscript was very carefully written.

The authors focused on the role of anthocyanins and anthocyanidins in osteoarthritis.
There was a detailed description of how the selection of publications (research) was made.
The figures/schemes presented in the paper are appropriately prepared. For the reader it is easy to understand.
In addition, in the discussion chapter, the authors have referred to a number of interesting papers, which appropriately enhanced the merit of this manuscript. The conclusion presented at the end is also adequately written. It is important that the authors have also addressed the limitations of their manuscript.

Author Response

(The authors gave the same response as above.)

Reviewer 4 Report

Comments and Suggestions for Authors

In this paper, the authors provided a valuable review of the role of anthocyanins and anthocyanidins in the management of OA. The paper is well-written and interesting. However, the following should be addressed to improve the quality of the manuscript:

Rephrase the following sentence for clarity:
"With a global age-standardised prevalence rate of the knee of 6,967.29 per 100,000 [95% uncertainty interval (UI): 6,180.7–7,686.06] in 2021 [1], osteoarthritis (OA), a disease characterised by cartilage degeneration, subchondral osteophyte formation, synovitis, and manifested as joint pain, swelling and stiffness [2], poises to impact the life of elderly significantly."

The terms in vitro and in vivo are not written in italic, they should be consistently italicized throughout the manuscript.

Rewrite the following sentence or use another word instead of "assaults": "Chondrocytes undergo apoptosis when exposed to assaults, such as IL-1β and oxidative stress"

The abbreviation for protocatechuic acid (PCA) should be consistently formatted throughout the manuscript.
The abbreviation IL-1β (interleukin-1β) should be introduced properly, and its formatting should remain consistent - "IL-1β" instead of "IL-1b" when referring to interleukin-1β (see Lines 146 and 213).

What could relieve the effects of obesity? Consider replacing "it" with the compound you are referring to: "Since obesity is a significant risk factor for OA due to increased mechanical loading on the joint, systemic inflammation and oxidative stress [51], it could relieve the effects of obesity on the joint."

Comments on the Quality of English Language

Minor corrections are required.

Author Response

(The authors gave the same response as above.)
